# *Yarrowia lipolytica* as an Alternative and Valuable Source of Nutritional and Bioactive Compounds for Humans

**DOI:** 10.3390/molecules27072300

**Published:** 2022-04-01

**Authors:** Monika Elżbieta Jach, Anna Malm

**Affiliations:** 1Department of Molecular Biology, The John Paul II Catholic University of Lublin, Konstantynów Street 1I, 20-708 Lublin, Poland; 2Department of Pharmaceutical Microbiology, Medical University of Lublin, Chodźki Street 1, 20-093 Lublin, Poland; anna.malm@umlub.pl

**Keywords:** amino acids, bioactive compounds, minerals, nutritional yeast, single cell oil, single cell protein, protein biomass, yeast protein production, vitamins, *Yarrowia lipolytica*

## Abstract

Yarrowia lipolytica, an oleagineous species of yeast, is a carrier of various important nutrients. The biomass of this yeast is an extensive source of protein, exogenous amino acids, bioavailable essenctial trace minerals, and lipid compounds as mainly unsaturated fatty acids. The biomass also contains B vitamins, including vitamin B12, and many other bioactive components. Therefore, Y. lipolytica biomass can be used in food supplements for humans as safe and nutritional additives for maintaining the homeostasis of the organism, including for vegans and vegetarians, athletes, people after recovery, and people at risk of B vitamin deficiencies.

## 1. Introduction

*Yarrowia lipolytica* is a well-known oleaginous yeast that has an ability to degrade wide range of hydrophobic substrates, effectively producing a huge number of valuable metabolic products such as protein and peptides, amino acids, trace minerals, vitamins, and carbohydrates or single-cell oil (SCO), predominantly as mono-unsaturated fatty acids (MUFAs) and saturated high-added-value lipids such as cocoa-butter equivalents [1,2,3,4,5,6,7,8,9,10,11,12,13,14]. For this reason, in recent years, it was the most studied unicellular fungus right after the conventional true yeast *Saccharomyces cerevisiae* [15,16]. The production of numerous metabolites by yeast cultured on agricultural, industrial, and food residues and wastes is perfect alternative way for the chemical and food industry. Moreover, this production helps to reduce environmental pollutions due to the conversion of waste and byproduct substrates to value-added products [3,8,9,17]. *Y. lipolytica* utilizing waste as a raw cassava (bioethanol industry by-product) reduces poisonous cyanide concentration to a safe level [18]. The yeast has relatively low nutritional requirements, showing very high growth potential, and its culture is independent of geographic and weather conditions and fresh water availability [6,19,20,21]. Additionally, the use of yeast biomass enriched in protein containing plenty of value compounds or its metabolites is easier accepted by society than other microorganisms due to its usefulness, i.e., in the production of fermented products [20].

*Y. lipolytica* strains are widespread in nature and can be easily isolated from various environments (i.e., water and soil) such as fat-rich substrates (i.e., oily wastes and sewage), hypersaline and marine sites, as well as contaminated soil at a car wash [22,23,24,25,26]. The yeast strains as a living cells also occur in wide ranges of food products with high amounts of fat and protein, e.g., dietary products such as cheese, especially blue cheese and cheddar cheese, milk, cream, kefir, yoghurt, butter, and margarine; as well as ripening meat products, e.g., various types of fermented sausages such as salami, and Spanish fermented sausages (i.e., chorizo); and various others, such as juice, cider, fruit, fruit concentrate, mayonnaise, seafood, raw plant materials, and wine [24,27,28,29]. Moreover, the yeast was isolated from a healthy human body. Therefore, *Y. lipolytica* is considered as normal human microbiota belonging to saprophytes in the adult respiratory system. It is also detected in breast tissue, some cutaneous nodules, and the digestive tract [30]. The natural occurrence of live yeast confirms their safety. The yeast is non-pathogenic for humans and animals. The live yeast cells in extreme cases can rarely cause an opportunistic fungemia exclusively in immunocompromised individuals with underlying diseases and those with catheters [24,27,31]. Thus, *Y. lipolytica* is considered as safe, and various production processes based on the yeast were classified as Generally Recognized as Safe (GRAS) by Food and Drug Administration (FDA) [27]. It is also classified as a Biosafety Level (BSL) 1 microorganism by the Public Health Service (Washington, DC, USA) as well as recognized as a “microorganism with a documented use in food” by the International Dairy Federation (IDF) and the European Food and Feed Cultures Association (EFFCA) [32].

In the 1950s in Great Britain, *Y. lipolytica* was used by British Petroleum Co. (BP) as a producer of single-cell protein (SCP) for animal feeding [27]. In 2010, throughout the European Union, dried and heat-killed biomass of *Y. lipolytica* cultivated in biofuel waste has been approved for animal nutrition [24]. Since then, a lot of clinical studies were carried out on *Y. lipolytica* biomass-fed animals such as fish species. e.g., Atlantic salmon [33,34], Nile tilapia [35], and Pacific red snapper [36]; birds, e.g., turkeys [37,38,39], piglets [40,41,42]; and ruminants, e.g., calves [34]. Dried and heat-killed yeast biomass work effectively as immunostimulants and animal growth promoters [35,37,39,43].

The growth of the human population leads to increased food production, especially protein-rich products. The need to provide food to so many people throughout the world creates a growing gap between suppliers and demand, mainly in developing countries [44,45]. Therefore, the search for new alternative ways of obtaining protein-enriched food is crucial to solving the increasing problem of food deficit. In the near future, the problem could be solved by the production of huge quantities of nutritional yeast biomass. At present, mainly S. cerevisiae biomass and extract are used for human nutrition but not commonly and in small numbers [19,46]. However, the first step to use the Y. lipolytica biomass as food has already been taken. In 2019, European Food and Safety Authority (EFSA) has allowed to use the biomass of Y. lipolytica cultured in waste substrate (i.e., biofuel waste) as a novel food in dietary supplements for the general human population above 3 years of age. The Y. lipolytica biomass was recognized as safe and nutritionally advantageous as shown in Figure 1. The proposed maximum daily portion is 3 g/day for children from 3 years up to 10 years of age and 6 g/day thereafter [24].

The aim of this review is to provide comprehensive information about nutritional metabolites, their genetic and enzymatic production machinery, and conditions for the utilization of wastes and by-products for the production of value compounds.

## 2. Characteristic of *Yarrowia lipolytica*

*Y. lipolytica*, formerly named *Candida lipolytica*, afterwards *Endomycopsis lipolytica,* and then as *Saccharomycopsis lipolytica*, just like *S. cerevisiae*, belongs to the kingdom of fungi, sub-kingdom *Dikaryota*, division *Ascomycota*, sub-division *Saccharomycotina*, phylum *Ascomycetes*, class *Saccharomycetes* (called *Hemiascomycetes* as well), and order *Saccharomycetales* [15,24,27,47,48]. *Y. lipolytica* belongs to the family *Dipodascaceae* and genus *Yarrowia*. The name was proposed by van der Walt and von Arx in 1980 in thanks for the identification of a new genus by David Yarrow [29,48]. The species name ‘lipolytica’ was given due to the ability of this yeast to hydrolyze fat substrates [29]. *Y. lipolytica* and *Candida lipolytica* are the teleomorth and anamorth (respectively) names of the same species [24]. *Y. lipolytica* belongs to the *Yarrowia* clade, a big group that includes many different species of yeasts from genera *Candida* and *Yarrowia*, such as *C. alimentaria*, *C. bentonensis*, *C. galii*, *C. hispaniensis*, *C. hollandica*, *C. oslonensis*, *C. parafonii*, *C. pseudorhagii*, *Y. brassicae*, *Y. bubula*, *Y. deformants*, *Y. divulgata*, *Y. keelungensis*, *Y. phangngaensis*, and *Y. porcina* and *Y. yakushimensis*, which were isolated from various biotic and abiotic environments (i.e., food, marine water, termite intestine) [49,50,51].

The genome sequence of *Y. lipolytica* shows that the yeast is distantly related to *S. cerevisiae*. The fundamental genetic mechanisms of *Y. lipolytica* are significantly different in comparison to *S. cerevisiae*. The genome of *Y. lipolytica* reveals an accretion of genes and protein families that are involved in hydrophobic substrate utilization, particularly fat-rich ones [52].

*Y. lipolytica*, as a unicellular form, multiplies vegetatively by polar budding, as in the conventional yeast form, and in sexual reproduction, forms asci which comprise 1–4 haploid ascospores [53]. *Y. lipolytica*, as a heterothallic microorganism, naturally occurs in two mating types, marked as MatA and MatB [29].

In contrast to *S. cerevisiae*, *Y. lipolytica* can grow as a dimorphic form, making it an ideal model for dimorphism studies. It can produce true filaments and presents pseudo-hyphae [27,29,54,55]. *Y. lipolytica* may form residual mycelium, well-developed mycelium, or does not form filaments at all. Not filamentous forms are usually oval and ellipsoidal. The creation of dimorphic forms depends primarily on adverse environmental conditions such as alteration in oxygen, pH, carbon, and nitrogen substrates [29,56]. Elongation into hyphae may be induced by replacing glucose with *N*-acetylglucosamine or by the addition of the serum to the medium [57]. Wild-type (wt) strains exhibit various colony morphologies that depend on culture conditions and genetic backgrounds. Colonies’ looks may vary from heavily convoluted and matte to smooth and glistening [29]. Moreover, *Y. lipolytica* is able to form a biofilm in the presence of different carbon sources, including alkanes [58]. The ability of *Y. lipolytica* to form biofilms may be important in natural environments, as well as when it grows in different wastes [15]. Wt *Y. lipolytica* utilizing various wastes or by-products (e.g., glycerol, alkanes, animal or vegetable fats, fatty esters, soap-stocks, pure free-fatty acids, petrol byproduct, and biofuel waste) can produce many bioavailable nutritional compounds [1,2,3,7,8,9,10,11,12,13,14]. Apart from glycerol and oily wastes, *Y. lipolytica* can degrade several hexoses, e.g., glucose, fructose, galactose, and mannose [59]. The yeast does not utilize sucrose and lactose [60]. However, one isolate from soil (*Y. lipolytica* B6) can degrade lactose [61]. The yeast also has the complete genetic pathway for xylose and arabinose utilization. Despite that, there are no studies to support the utilization of these substances by the yeast [62]. When glucose or glycerol are absent, *Y. lipolytica* can utilize ethanol, acetic, butyric, citric, lactic, malic, propionic, and succinic acids as carbon and energy sources [54,63,64]. Interestingly, lactic acid may be also used as a substrate when nitrogen is unavailable (i.e., free amino acids) in the culture medium [65]. The wt yeast does not synthesize carotenoid dyes and does not assimilate inositol and nitrates [20].

## 3. Culture Condition in Various Wastes for Nutritional Biomass Production

*Y. lipolytica* requires strictly aerobic condition to grow. Meeting this condition is actually the only difficulty in growing this yeast in an industrial scale. *Y. lipolytica* can be cultured at a wide range of temperatures from 2 °C to 32 °C and pH from 2.5 to 8.0 [56]. Most *Yarrowia* strains are incapable of growing above 32 °C, which supports the thesis about their safety of use for human nutrition [29]. It is considered that its optimal culture conditions are ambient temperatures between 28 °C and 30 °C and mild acidic pH about 5.5 [66]. However, as we showed in Table 1, depending on the waste substrate used, the optimal pH value changed. These pH values were in range from strong acidic (3.0) to mildly acidic (5.5). Very low pH may prevent the yeast culture from contamination with other microorganisms. These productions take a relatively short time, between 12 h and 120 h, depending on culture conditions and waste medium used.

Apart from temperature and pH, important parameters for the fermentation process are also the aeration rate, dissolved oxygen, or the moisture content of solid medium [78]. The choice of waste substrate and all culture conditions must be based on optimization studies, preferably using statistical techniques [9,68,79,80,81,82,83]. For the production of protein-enriched biomass, presented species (Table 1) were cultured both in liquid and on solid waste materials, mainly industrial substrates such as glycerol, biofuel waste, and solid tallow, as well as agricultural wastes such as rye straw, ray and oat brans, and soy bean hydrolysates. In case of substrates containing fiber, e.g., inulin or Jerusalem artichoke tubers, it was necessary to use recombinant strains (mutants) due to wt *Y. lipolytica* not being able to produce cellulases and hemicellulases; thus, it does not utilize fiber in agricultural or forestry residues [84]. In this regard, it is postulated to improve obtaining nutritional biomass by developing genetic engineering procedures for mass production [85]. However, there is a big problem due to the strict regulation on the use and culture of genetic modified organisms (GMO) in the human diet in the European Union [72]. Thus, for large-scale yeast biomass production at competitive prices constituting a feasible replacement for soybean and fishmeal protein in nutrition, including aquaculture [86], wt yeast culture development is strongly desired.

The capacity of growing wt strains in a wide-range of wastes contributes to the biodegradation of the wastes, especially alkanes, petroleum by-products, glycerol, biofuel waste, and plant or animal-waste fats [3,6,8,9,14,17]. Furthermore, liquid substrates are dispersed more easily with moderate agitation than solid media (i.e., tallow), which require strong agitation (i.e., 1200 rpm) for dispersal in the culture materials [75]. It is worth emphasizing that the oily waste biodegradation by the yeast is essential for environmental preservation, in line with the take–make–dispose concept [9,87,88,89]. Moreover, as for being environmentally friendly, *Y. lipolytica* tolerates heavy metal ions, including cadmium, cobalt, and nickel, as well as high salt concentrations up to 12% (*v*/*v*) [56]. Thus, it is involved in the detoxification of heavy metals and other pollutants [15].

Sometimes, during the optimization process, there is a need to add into the waste medium various nutrients such as nitrogen, phosphorus, and others to improve optimal yeast growth and value bioactive compound production by selected strain [90]. The presence of nitrogen in the medium has a fundamental impact on the metabolic pathways and the production of metabolites [65]. Nitrogen availability is particularly important for protein synthesis [91]. In turn, nitrogen deficiency in the medium is desirable for citric acid production or lipid accumulation as storage lipid bodies when glucose or glycerol are in the culture medium. However, there are *Y. lipolytica* strains which may accumulate lipids when nitrogen is available or just after being deprived of nitrogen. Then, when nitrogen-limited conditions were reached, the strains consumed the storage lipid bodies [92].

Interestingly, the addition of selenium (NaHSeO_3_) from 1 to 6 mg/dm^3^ lowered the amount of protein in biomass of *Y. lipolytica* A-101 cultured in raw rapeseed oil twofold (from 56.4 to 24.7%, respectively). In turn, the addition of chromium (CrCl_3_ × 6H_2_O) from 5 to 80 mg/dm^3^ into the same medium caused a decrease in protein content from 39.4 to 20%, respectively [71]. The addition of 4 mg/dm^3^ selenium (IV) and 40 mg/dm^3^ chromium (III) in the culture medium was the most effective at simultaneously producing biomass rich in protein and high concentrations of these minerals. These results were confirmed by Juszczyk et al. [70]. These studies showed [70,71] that regardless of the carbon source, *Y. lipolytica* is characterized by a good accumulation of selenium and chromium in the cell.

The search for the best bioproduct producer should also be based on optimization methods. One of the most effective methods to derive new highly productive strains is an adaptive laboratory evolution (ALE) strategy. Daskalaki et al. [93], using ALE, obtained the strain that accumulated 30% more fats after 77 generations than the starting *Y. lipolytica* strain.

In case of yeast biomass production for human consumption, a process step should be added for the reduction or removal of nucleic acid to a safe level of 1% because higher RNA concentrations when consumed by humans can be toxic [94]. The biomass of most yeast contains low amounts of nucleic acid in ranges from 5% to 8%. It is lower than other microorganisms contain, e.g., bacteria contain 8 to 15% [95]. For example, the biomass of *Y. lipolytica* cultivated in biofuel waste contained from 6.4% of RNA in an initial lag phase, through 5.8% in the end of log phase to 0.4% at the end of stationary phase [6]. It was observed earlier that during stationary phase microbial cells have the lowest content of nucleic acids, which is partially caused by reduction of RNA quantity [96]. Hence, one of the most effective methods of nucleic acid reduction is simple elongation of *Y. lipolytica* culture time up to the final stage of the stationary phase while endogenous nucleases (ribonucleases) are activated [6]. Endogenous nucleases are also activated by thermal shock (60–90 °C) [97]. Other less recommendable methods are the following: chemical treatment with NaOH or 10% NaCl, the addition of ribonucleases to the cultivation process, or the use of these nucleases as immobilized enzymes [97,98,99] and two-step treatment with a chemical combination of *N*-lauroyl sarcosine and NH_4_OH, which reduced RNA to below 2% [100].

To obtain edible, digestible and bioavailable *Y. lipolytica* cells, they should be destroyed by drying at high temperatures, resulting in dried and heat-killed biomass adequate for human consumption [6,24]. Among cell disintegration methods (e.g., crushing, crumbling, grinding, pressure homogenization, or ultrasonification), drying at high temperature is one of the most effective methods of preventing contamination and extending the shelf life of feed or food [101]. The moisture content of dried *Yarrowia* powder was lowered below 6% [6,24]. A moisture level below 8% is considered as safe for long shelf life of products and helps to achieve proper texture and stability of yeast biomass as food ingredients [102]. Additionally, drying at high temperatures (e.g., 165–175 °C) kills cells, disintegrates the cell wall, and contributes to the release of the yeast cells’ contents. Therefore, drying significantly improves the quality of yeast biomass [46,85]. A heat-killing step of yeast biomass production results in the absence of viable *Y. lipolytica* in the powder [103,104]. Dried and heat-killed *Y. lipolytica* biomass obtained after the culture in biofuel waste is an amorphous hygroscopic beige-colored powder, as shown in Figure 2, with a slight yeast odor [5,40].

## 4. *Y. lipolytica* as a Producer of Protein Biomass

### 4.1. Protein Concentration

*Y. lipolytica* cultivated in varied oily substrates produces plenty of nutritional compounds; therefore, it was given a name nutritional yeast. As shown in Table 1, *Y. lipolytica* is an excellent producer of protein such as SCP. The *Y. lipolytica* biomass contains abundant protein in range between 30.5% and 56.4%, with an average protein value of 43%. It is considered that the preferable protein concentration is just above 40% [105]. EFSA and FAO recommend an intake of 50 g of protein in daily human diet for adults [106,107]. Hence, 100 g of *Y. lipolytica* powder contains almost 100% of the recommended daily portion. Moreover, protein quantity in *Y. lipolytica* powder is comparable to that of *S. cerevisiae* biomass and is similar or even higher than in traditional sources such as meat and soybean and higher than milk protein [19,46,72]. Furthermore, the protein efficiency ratio standardized for casein (PER) of yeast protein is comparable with the PER of meat and soybean [72,108]. The digestibility coefficients of *Y. lipolytica* protein are very high and are in the range of 72.3% and 77.2% [70,72,74] and are comparable to that of *S. cerevisiae* (73.7%) [72]. Because *Y. lipolytica* has a comparable chemical composition to that of *S. cerevisiae*, it may to have similar functions in human and animal organisms [38]. Moreover, the digestibility coefficients of protein biomass for both yeast species are similar in the feeding of different animals such as rats and piglets [109]. In turn, the digestibility of the ether extract of *Y. lipolytica* grown on industrial glycerol was 57.3%, as shown in Table 2, while, for *S. cerevisiae*, the ether extract was absent entirely [72].

The protein biomass of *Y. lipolytica* cultivated in industrial glycerol or biofuel waste contains good portion of dietary fiber up to 30.7% of total dry matter (Table 2) [24,72,103,104]. *Y. lipolytica* fiber is mainly composed of β-glucans, complex polysaccharides that form part of the cell walls of yeasts and cereals [24,103,104,111]. In the Western world, human diets are scarce in fiber. Recently, Barber et al. [112] analyzed many studies from which they summarized that dietary fiber intake is associated with overall metabolic health (through key pathways that include insulin sensitivity) and a variety of other pathological conditions including cardiovascular disease, colonic health, gut motility, and colorectal carcinoma risk. Dietary fiber intake is also correlated with mortality. The gut microflora works as a significant mediator of the pro-health effects of dietary fiber, including the regulation of appetite, metabolic processes, and chronic inflammatory pathways; hence, β-glucans are considered as prebiotic.

### 4.2. Amino Acid Content in Y. lipolytica Protein Biomass

The protein nutritional value is determined by proper amino acid composition, predominately of eight essential amino acids which the human body is incapable of synthesizing [97]. As shown in Table 3, *Y. lipolytica* powder contains a complete set of amino acids, making this biomass a valuable source of complete protein [6,67,70,72,113,114,115,116]. It is worth emphasizing that *Y. lipolytica* biomass enriched in good-quality protein is obtained in a much shorter time and much easier and cheaper, with a low environmental footprint and an environmentally friendly process that utilizes various wastes, which traditional protein sources such as plants and animals do not.

The number of amino acids depends on the medium used. The lowest amino acid levels were obtained when the yeast was grown on rye straw and rye or oat brans [67]. However, the use of fatty substrates such as crude or industrial glycerol, or biofuel waste enhanced the number of amino acids resulting in much higher quantities, yielding them in the upper limit (Table 3) [6,68,72]. The essential amino acid profile in the protein of *Y. lipolytica* when grown in these fatty substrates (i.e., glycerol and biofuel waste) is comparable to or higher than those required by the FAO for daily human diet (Table 3). It is very important because protein deficiency may occur even when one of the essential amino acids is missing [117]. Moreover, the protein of the yeast cultured in the fatty materials such as glycerol and biofuel waste contain high quantities of lysine, of which there is too little in wheats, including cereals [118].

### 4.3. Genetic and Enzymatic Machinery of Protein Biomass Production

It is known that *Y. lipolytica* growing in oily substrates accumulates huge amounts of lipids intracellularly as storage lipid bodies (more than 40%), referred to as single-cell oils (SCOs) [1,119,120]. However, the production de novo and the accumulation ex novo of SCOs depends on the type of strains and the culture substrates, and the two processes can occur simultaneously [121]. Wt strains, cultured in sugar medium, produce lipids less efficiently, reaching 4–20% of intracellular lipids in dry weight (DW), or more efficiently when they are cultivated in hydrophobic materials [8,10,56,121,122]. Production referred to as de novo occurs when the yeast grows in sugar medium and similarly metabolized substrates, i.e., glycerol. SCOs are synthetized during a secondary metabolism, which occurs especially when nitrogen is depleted or a high carbon/nitrogen (C/N) ratio in the medium exists [8,10]. Nitrogen starvation in the environment starts a cascade of metabolic reactions, leading to a rapid reduction in intracellular AMP, which causes mitochondrial AMP-activated isocitrate dehydrogenase (ICDH) inhibition. ICDH inhibition is decisive in signaling the first step of the lipogenesis process due to the disturbance of the Krebs cycle, inducing the strong accumulation of intramitochondrial citric acid, which is excreted to the cytoplasm instead of malate. Then, in the cytoplasm, citric acid is hydrolyzed to acetyl-CoA and oxaloacetate by ATP citrate lyase (acyltransferases, ACL). It is noteworthy that the transcription of genes for ACL and ICDH is observed in both oleaginous and non-oleaginous conditions [1,3]. Moreover, for even more effective lipid accumulation by *Y. lipolytica*, apart from nitrogen starvation, magnesium starvation was also favored with comparison to the presence of only one of the two (from 13.6 to 38.5%). Although, for effective lipid production, it is sufficient to provide nitrogen starvation or high C/N ration [1,56]. However, it is detected that the yeast has already produced lipids in the first step of its growth when both nitrogen and crude glycerol as carbon and energy source in the medium were presented [92,123]. Furthermore, simple pH control could also increase lipid accumulation [124]. In turn, ex novo lipid production as the primary anabolic cell process occurs when the yeast is cultured in fat-rich substrates. When *Y. lipolytica* is grown in fatty materials, it is proven that the yeast rapidly incorporates unsaturated fatty acids, i.e., oleic acid for proper growth and organic acid production. However, saturated fatty acids (SFAs), e.g., stearic acid, are slowly incorporated and used for proper growth, as well as SCO production [10]. SCOs are primarily composed as triacylglycerols (TAGs, called neutral fats, triglycerides TGs, or triacyl glycerides) with low levels of free fatty acids, neutral lipids, polar fractions, and sterols. Among the accumulated fatty acids, oleic, linoleic, and palmitic acids are mostly represented [66,125,126]. Moreover, the modulation of growth parameters and substrates resulted in obtaining tailor-made SCOs [8,10]. It is observed that *Y. lipolytica* cultured in various oils or oily wastes produced SCOs abundant in oleic acid. In turn, the yeast cultured in SFAs, e.g., stearin, generated SCOs rich in SFAs. Furthermore, when the yeast is grown in a mixture of hydrolyzed rapeseed oil and stearin in proportion 50:50, its lipids were composed similarly to cocoa butter [10].

However, it is proven that ex novo lipid production is independent of nitrogen starvation. Utilizing accumulated SCOs, fat-free metabolites are produced [10,56]. Hence, the protein biomass of *Y. lipolytica* grown in agricultural (i.e., rye and oat straw or brans) or industrial wastes (i.e., industrial or raw glycerol) contains low concentrations of total cellular lipids (8.2–20%, respectively), with a high spectrum of unsaturated fatty acids (65–90% of total lipids). Among the unsaturated fatty acids, it is dominated by MUFAs (57.6%), represented by oleic acid (34–59.2%), and in low concentrations of poly-unsaturated fatty acids (PUFAs) (34%), represented by linoleic acid (11.9–27%). Protein biomass has low amount of SFAs (8.3–34.9% of total lipids) [67,72]. The low content of total lipids in protein biomass is the result of using accumulated lipids in storage bodies as source of carbon and energy for the yeast cell growth when the extracellular carbon source is depleted and protein or cellular carbohydrates biosynthesis. Thus, the accumulation of lipids and protein or carbohydrates production are competitive processes [3,93].

The degradation of hydrophobic materials (i.e., alkanes, fatty acids, and oily wastes) by *Y. lipolytica* is a complicated multistep process in which a number of genes encoding protein family enzymes responsible for the fat utilization are involved [29]. *Y. lipolytica* possesses more than 25 genes encoding lipases and estreases, among them, *LIP2*, *LIP7*, and *LIP8* are specific for fatty acid chains of lengths C18, C6, and C10, respectively. Extracellularly released lipase (Lip2) has a high specificity towards saturated triglyceride-tricaprylin (C8:0) and TGs containing MUFA triolein (C18:1), as well as C12, C14, and C16 methyl esters [127,128,129]. *Y. lipolytica* also produces intracellular lipases (Lip1, Lip3, and Lip6) and cell-bounded Lip7 and Lip8 [127,128]. The latter lipases prefer *p*-nitrophenyl C8–C12 esters and *p*-nitrophenyl C8–C10 ones [129]. Additionally, the yeast has six *POX* genes which are characteristic only for oleaginous ones. Among them, *POX*2 and *POX*3 code long and short specific acyl-CoA oxidase enzymes, respectively, which directly allows the yeast to utilize fat-rich substrates by complete oxidation using intermediates of β-oxidation [59,130,131,132]. Interestingly, a transfer of the one gene encoding acetyl-CoA oxidase from *Y. lipolytica* to *S. cerevisiae* enabled *S. cerevisiae* to grow in fat-rich medium [133]. Furthermore, a knockout of the all-*POX* genes in *Y. lipolytica* causes the inability of this yeast to degrade storage lipids, leading to an over-accumulation of fats in yeast cells [59,134]. Moreover, *Y. lipolytica* possesses 12 genes from the CYP52 family for cytochrome P450. Some of them code specific enzymes for alkanes or fatty acid hydroxylation [29,52,135].

When the yeast is cultivated in fatty substrates, at first, the hydrophobic cell wall of *Y. lipolytica* secrets emulsifiers, surfactants, and extracellular lipases that hydrolyze TAGs to fatty acids and glycerol [25]. During the growth on fatty-acid-containing media, extensive proliferation of peroxisomes and intense accumulation of lipids as storage lipid bodies occur. In turn, when yeast grow on a fatty acid-poor or fat-fee carbon source (i.e., sugar medium), peroximal structures cannot be identified. Peroxisomes play the sole role in β-oxidation in the fatty acid degradation process causing the formation of acetyl-CoA in the peroxisome [136,137]. The SCOs accumulation lasts until extracellular carbon sources exist. Furthermore, extracellular lipase content decreases with depletion of the oily substrate in the stationary growth phase [75]. However, the addition of oily materials (i.e., waste cooking oil) into the culture medium induces extracellular lipase production by *Y. lipolytica,* but in fat-free cultures, it does not [9,138].

Since carbon sources in the culture medium are exhausted, the yeast have to utilize their own storage lipids as a carbon and energy source to survive and maintain vital functions, using intracellular lipases to hydrolase SCOs, then consequently decreasing fat content in the yeast cells and increasing the production of protein and other fat-free metabolites by shifting the metabolism from SCOs production to fat-free metabolites biosynthesis [1,3,10,56,75,93,139,140]. This process leads to obtaining de-fatted *Y. lipolytica* biomass containing half of its weight as protein (up to 56.4%) with all the essential amino acids in significant amounts. It has been proven that with the increase in protein in the yeast biomass, there is a significant degradation of the storage lipid bodies dependent on the availability of nitrogen and magnesium in the culture medium [1,93]. Moreover, it was observed that the enhancement of the state of aeration and agitation of the culture medium increased the activity of short-chain acyl-CoA oxidases. Consequently, the carbon flow was driven to the synthesis of acetyl-CoA and then to formation of de-fatted protein biomass rather than storage lipophilic biomass [141]. Of note, both extracellular and intracellular lipases show their activity in a wide spectrum of temperatures (4–55 °C), depending on the used strain [127,129,141]. However, the maximum activity of the lipases was found at the temperatures between 30 °C and 40 °C and at pH 5.0, while a significant lowering in their stability was observed at pH values over 6.5 [9,142]. Moreover, the most efficient production of protein biomass with many bioactive compounds was also noted under similar conditions: temperatures of 30 °C and pH 5.0 [6,68]. Therefore, the manipulation of culture conditions (i.e., agitation and aeration), oily medium compounds (i.e., addition of nitrogen and magnesium), and the depletion of the external carbon source collectively contribute to production of the de-fatted protein-rich biomass.

### 4.4. Concentration of Trace Minerals and Vitamins in Y. lipolytica Protein Biomass

*Y. lipolytica* protein biomass is a good source of macro-elements such as calcium, copper, phosphorus, potassium, and zinc and micro-elements such as chromium, iodine, and selenium, as shown in Table 4. Those elements are in bioavailable organic forms found in the biomass [19,24,38,46,69,70,71,72].

In a similar way to *S. cerevisiae*, *Y. lipolytica* is capable of assimilating minerals from the culture medium and incorporating them into its cell organic structures [40,66,69]. For example, chromium as a trace element is essential for the proper metabolism of glucose, insulin, fatty acids, proteins, and muscle growth. It is suspected that its adequate supply may reduce the fat content of pig carcasses. Minerals in yeast protein biomass, such as chromium or selenium, behave like chelates in the animal’s body. As a result of feeding organic forms with micronutrients, their absorption and use is better, which is of great physiological and ecological importance and reduces environmental contamination [69]. After a three-step sequential extraction (i.e., fractionation) and a simulated in vitro gastrointestinal digestion, it has been detected that in *Y. lipolytica* protein biomass, chromium occurred in the water-soluble part (50.2% of total Cr) and as polysaccharide-bound fraction of chromium (43.9% of total Cr) and protein-bound fraction (4.3% of total Cr). Moreover, the whole amount of Cr was trivalent (Cr (III)). The toxic form of Cr (IV) was not detected [104]. For human nutrition, according to EFSA recommendation [104], the maximum total *Y. lipolytica* chromium content of the product should be 23 μg/g, with the micro-nutrient present as Cr (III). The target population is general population from 3 years of age upwards, with maximum proposed use levels of 2 g/day for children from 3 to 9 years of age and 4 g/day thereafter. EFSA considered that chromium-enriched biomass of *Y. lipolytica* is safe under proposed condition of use.

In case of selenium from *Y. lipolytica* protein biomass, according to EFSA recommendation [104], the maximum total Se content in the products for children from 3 to 9 years of age is 40 μg Se/day and 200 μg Se/day thereafter. Se in the protein biomass is primarily presented as bioavailable organic selenium compounds: L-selenomethionine and L-selenocysteine, with low amounts in inorganic compounds as selenate and selenite [143]. It is worth emphasizing that the Se compounds provided by *Y. lipolytica* protein biomass are the same and as safe as those present in other dietary sources [103]. An advantageous feature of *Y. lipolytica* protein biomass is that it has low concentrations of sodium [40,69].

The protein biomass of the yeast is also a good source of vitamins, especially B-complex group. As shown in Table 5, protein biomass of *Y. lipolytica* cultured in industrial glycerol or biofuel waste contains a good amount of B vitamins, such as thiamine (vitamin B1), riboflavin (vitamin B2), pyridoxine (vitamin B6), folic acid (vitamin B9), and cyanocobalamin (vitamin B12) [4,5].

However, biotin (vitamin B7) may be produced in insufficient quantity (mean 20 μg biotin/100 g DW) by *Y. lipolytica* when cultivated in biofuel waste [5]. The yeast is able to synthesize the vitamins de novo or accumulate them ex novo (e.g., vitamin B12) from the aqueous environment in a similar manner to animal cells [4,5,70].

The bioavailability of dietary B vitamin-enriched *Y. lipolytica* biomass is very high [144,145]. These vitamins are water-soluble, and a surplus of them is generally excreted in urine [146]. Therefore, their doses must be much higher than the nutrient-recommended values (NRVs) [110] and, simultaneously, more consistent than that of the lipid-soluble vitamins. Interestingly, in developed countries, diets are impoverished of B vitamins [146]. The quantities of B vitamins in 100 g of the *Y. lipolytica* protein biomass completely cover the required NRVs for vitamin B1, B2, B6, B9, and B12. Hence, biomass enriched in B vitamins should be included in thediet for people who avoid eating meat and other animal products, particularly vegans and vegetarians, or those whose diet is poor in animal products. Deficiency of B vitamins, especially vitamin B12, leads to serious health problems [146,147,148,149,150].

## 5. Added-Value Compounds for Industrial Application

*Y. lipolytica* is considered as a ‘workhorse’ of a wide range of biotechnological production [54]. When the yeast is grown in various wastes and by-products, it is used as a producer of fatty acids [3,55,151]. It also produces lactones used as “fruity” aroma. Among them, the most interesting one is γ-decalactone, which is produced by the yeast when cultured in ricinoleic acid [152].

The yeast growing in various substrates under different conditions also produced some organic acids such as citric acid and isocitric acids [153,154,155,156,157,158,159,160,161,162,163,164], α-ketoglutaric acid (KGA) and pyruvic acid (PA) [27,56,165,166,167,168], and kynurenic acid (KYNA) [169]. Citric acid is commonly used as a flavoring agent, acidifier, antioxidant, and preservative in the food and pharmaceutical industries [140,170]. Of note, in wt strains, citric and isocitric acids are usually co-produced. However, the co-production of these acids is frequently not desired due to difficulties in their separation and receiving pure compounds. Therefore, some researchers tried to modulate citric acid/isocitric acid ratio either via the modulation of culture parameters such as substrate composition, micronutrients or pH, or mutant creation [167,171,172,173,174]. Similarly, KGA and PA are co-produced and realized in specific culture conditions, such as thiamine deficiency, low pH, and substrate degradation by glycolysis, impairing Krebs’s cycle. By contrast, when the yeast cultured in oily substrates (e.g., rapeseed oil), PA is not accumulated [168]. Furthermore, the selection of wt strains or the alteration of culture conditions were intensively assessed for obtaining KGA without PA [165,167]. KGA and PA are important in food, pharmaceutical, fodder, and other industries [175,176]. *Y. lipolytica* cultivated on honey also produces a tryptophan metabolite, KYNA, which exhibits pro-healthy properties due to its antioxidant, anti-inflammatory, and anticonvulsant action. KYNA also can protect the human brain and an inhibit the proliferation of colon cancer cells (i.e., adenocarcinoma HT-29 cells) [169].

*Y. lipolytica* is able to produce some polyols such as erythritol and mannitol commonly used as food additives for their flavor enhancer, sweetener, and humectant features [177,178,179,180]. The yeast polyols are produced in conditions such as high C/N ratio, high sugars or similarly metabolized compounds, low pH (3.0–3.5), and low oxygen [181]. In the conditions depending on strain used, it mainly produced mannitol in amount of max 30 g/L (around 80–88% of total polyols) [92,182]. However, when the yeast was cultured in glycerol, increasing pH to 5.5 for 72 h caused a shift in the yeast cell metabolism towards citric acid production (up to 40 g/L). Moreover, there are isolated strains which can produce both citric acid and polyols [182]. However, *Y. lipolytica* recombinant (MK1) produced up to 225 g of erythritol/L from glycerol [183]. Mutants generally can produce more polyols than wt strains [173,183,184,185,186], but recombinants have weaker adaptive properties to environmental changes in the culture medium than wt strains. When NaCl content was changed in the culture medium, the wt isolate produced higher mannitol concentration than the mutant strain (14 g/L and 4 g/L, respectively) [179,181,187]. Interestingly, it was noted that the produced polyols were re-utilized completely by the yeast as energy sources for growth needs when glycerol was exhausted [188].

*Y. lipolytica* is a graceful model to genetically modify to improve its metabolic productivity, increase biomass yields, and/or expand the substrates to be utilized [56,62,179,189]. Metabolic engineering of *Y. lipolytica* to produce bioactive compounds seems to be a good choice, especially when the yeast is employed to utilize common forestry and agricultural wastes such as cellulose, inulin, starch, or xylan, unused by wt strains [77,190], and/or to produce novel compounds that are not produced by the wt yeast. Engineered *Y. lipolytica* strains produce several heterologous valuable metabolites such as carotenoids [191,192]; terpenes, e.g., limonene [193]; polyketides with various biological activities; and aromatic amino-acid-derived molecules [194,195].

*Y. lipolytica* also has an operational secretory system with which it releases huge quantities of enzymes, such as lipases [196] and proteases [197], and heterologous proteins [27,198,199,200,201,202] for industrial applications. Among the more than 100 recombinant proteins produced by engineered *Y. lipolytica* [202], there are several pharmaceutically important human-compatible bio-compounds secreted by ‘humanized’ strains, including interferon alpha 2b [203], epidermal growth factor [204], *N*-glycoproteins [205,206], blood coagulation factor XIIIa [207], proinsulin and insulinotropin [208], cytochrome P450 [209], estrogen receptor α [210], and anaphylatoxin C5a that possesses both spasmogenic and leukocyte-related properties [211].

## 6. Safety of *Y. lipolytica* as a Valuable Source of Bioactive Compounds for Food

Dried and killed *Y. lipolytica* protein biomass is recognized as safe for human and animal nutrition in accordance with the current food and nutrition safety law. The protein biomass of *Y. lipolytica* cultivated in wastes such as industrial glycerol or biofuel waste did not contain excessive quantities of toxic minerals such as arsenic, cadmium, lead, and mercury. The contents of heavy metals were in low levels and did not exceed the standards [24,72,103,104]. The other parameters, including the levels of pathogens and total amount of contaminated microorganisms (i.e., mesophilic bacteria, yeast, and mold) and pesticides such as organochlorinated and organophosphate pesticides, pyrethroids, and other pesticides, were also below their limits of quantification [24,103,104].

It is worth emphasizing that *Y. lipolytica* is naturally occurring in foods and that it is not known to cause allergic reactions in humans [24,103]. Moreover, *Y. lipolytica* is not among the yeast species which have been noted to elicit allergic reactions in humans [212]. EFSA considered that given the protein content of the *Y. lipolytica* protein biomass (45–55 g protein/100 g DW), allergic reaction to the biomass cannot be excluded. However, the risk of allergic reactions is low [24,103,104].

Additionally, it is recognized that *Y. lipolytica* protein biomass is gluten-free. It is proven that the nutritional yeast products, i.e., the protein biomass, contained gluten at below the 20 mg/kg threshold defined by the Codex Alimentarius, The Codex, or the Gluten-Free Certification Organization (GFCO) [213].

## 7. Conclusions

*Y. lipolytica* biomass is a valuable source of bioactive compounds such as protein with an appropriate spectrum of amino acids and low amounts of fats and sodium. The protein biomass is also rich in trace minerals; vitamins, especially B-group vitamins; and other valuable compounds. The nutritional protein biomass may be prepared as dietary supplements in the forms of capsules, tablets, and sachets or be used as food additive to improve nutritional quality food and feed. The use of the nutritional protein biomass is an excellent alternative for traditional nutrition sources such as plants and animals. Products from protein biomass are especially intended for people at risk of vitamin B deficiency who avoid animal products, including vegans and vegetarians, and those with low intake of animal foods, such as populations who do not consume animal products due to culture, conviction, poverty, or those with restrictive diet patterns, as well as athletes and people after recovery. These protein biomasses are obtained in a short time with the use of various wastes in a small land area; throughout the whole year and with a low environmental footprint; and independently from climatic and weather conditions and fresh water availability. Moreover, engineering *Y. lipolytica* creates new perspectives in terms of its use for the production of novel bioactive compounds, especially recombinant therapeutic proteins.

## Figures and Tables

**Figure 1 molecules-27-02300-f001:**
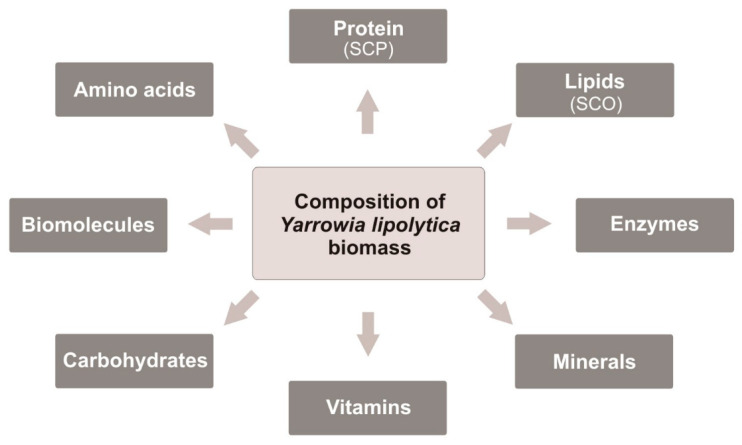
Composition of *Yarrowia lipolytica* biomass.

**Figure 2 molecules-27-02300-f002:**
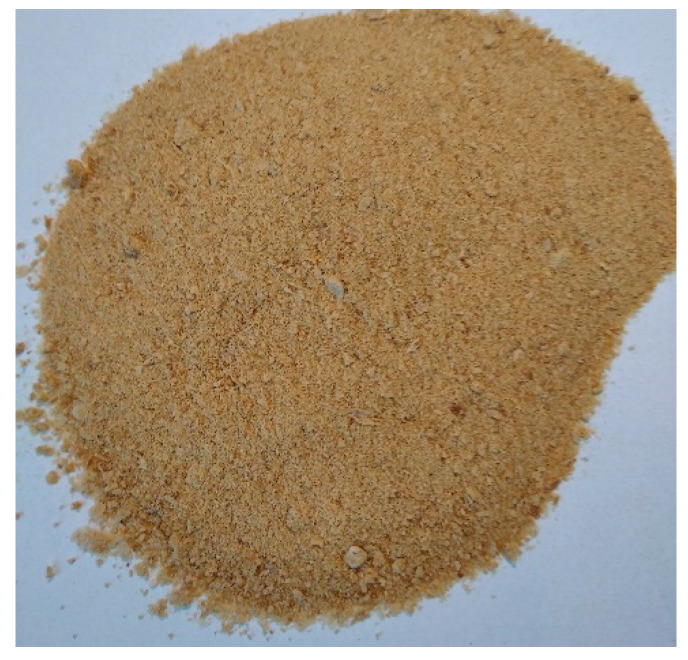
The dried powder of *Y. lipolytica*.

**Table 1 molecules-27-02300-t001:** Reports of *Yarrowia lipolytica* strains cultured in specific wastes, culture conditions, and protein content.

*Yarrowia lipolytica* Strains	Waste Substrates	Culture Conditions	Protein Content	Reference
wt A-101	Rye straw	Shake-flask, 28 °C, pH 3.5, with shaking at 150 rpm, 120 h	30.5%	[67]
	Biofuel waste	Bioreactor 100 L, 30 °C, pH 5.0, 40% oxygen agitation, 12 h	40–50%	[6,68]
	Rye brans	Shake-flask, 28 °C, pH 3.5, with shaking at 150 rpm, 120 h	44.5%	[67]
	Oat brans	Shake-flask, 28 °C, pH 3.5, at shaking 150 rpm, 120 h	44.5%	[67]
	Industrial raw oil waste with glycerol (88%)	Nd	45%	[69]
	Waste glycerol (78%) with degumming from rapeseed oil	Biostat B-Plus 5 L with mechanical agitation and bubble bioreactor 2000 L, 30 °C, pH 3.5–3.8	46.4–50.3%	[70]
	Raw rapeseed oil	AK 210 bioreactor, 30 °C, pH 3.5, with shaking at 750 rpm, 1 vvn oxygenation, 12 h	56.4%	[71]
Nd	Industrial glycerol	AK 210 bioreactor, 28 °C, pH 5.5, 1.2 L/ min of aeration, 48 h	46.7%	[72]
wt SWJ-1b	Soy bean cake hydrolysate with 20 g/L of glucose	28 °C, pH 5.5	41–47.6%	[73]
wt S6	pure and raw glycerol (25 g/L)	Bioreactor, 30 °C, pH 3.5	45%	[74]
wt S11	pure and raw glycerol (25 g/L)	Bioreactor, 30 °C, pH 3.5	50.1%	[74]
wt ACA-DC 50109	Industrial derivative of tallow (stearin 100%)	Bioreactor, 28 °C, pH 6.0, 650 rpm agitation, 0.3 vvm of aeration, 24 h	Nd	[75]
Recombinant 29a	Hydrolysate of soy bean meal with ammonium sulfate	Shake-flask 250 mL, 28 °C, at shaking 170 rpm, 120 h	46.1%	[76]
Recombinant	Inulin	Bioreactor, 72 h	47.5%	[77]
	meal of Jerusalem artichoke tuber (8.0%)	2–1 bioreactor, 80 h	53.7	

Nd—not disclosed; wt—wild type.

**Table 2 molecules-27-02300-t002:** Nutritional analysis of *Yarrowia lipolytica* protein powder (dry weight).

Parameter	Amount	Requirements for Adults [110]
Dietary fiber (g/100 g)	20.0–30.7	-
Ether extract (g/100 g)	57.3	-
Nitrogen-free extract (%)	96.2	-
Saccharides (g/100 g)	0–2.0	90 g
Total ash (%)	9.0–15.0	-
Dry matter content (%)	86.2–98.5	-

**Table 3 molecules-27-02300-t003:** Amino acids concentration in protein of *Yarrowia lipolytica* powder (dry weight).

Parameter	Amount	FAO Requirements for Adults
Protein (g/100 g DW)	20–56.4	50
Amino acid	mg/1 g protein	
Alanine	17.8–80.0	-
Arginine	12.7–48.0	-
Asparagine	31.8–88.0	-
Cysteine	3.5–11.0	-
Glycine	14.4–46.0	-
Glutamine	40.2–120.0	-
Histidine *	8.5–26.0	15
Isoleucine *	13.9–44.0	30
Leucine *	33.8–71.2	59
Lysine *	18.3–70.0	45
Methionine *	4.7–14.4	-
ΣSAA	8.2–25.4	22
Phenylalanine *	13.8–69.0	-
Proline	13.0–42.0	-
Serine	15.6–44.7	-
Tryptophan *	3.5–47.0	-
Tyrosine	8.1–110.0	-
ΣAAA	25.4–197.0	38
Threonine *	18.9–50.0	23
Valine *	17.7–53.3	39
ΣEAA	133.1–413.9	-

DW—dry weight; *—essential amino acid; ΣAAA—sum of aromatic amino acids: phenylalanine, tyrosine and tryptophan; ΣEAA—sum of essential amino acids; FAO—Food and Agricultural Organization; ΣSAA—sum of sulfur amino acids: cysteine and methionine.

**Table 4 molecules-27-02300-t004:** Trace mineral content in *Yarrowia lipolytica* protein powder (dry weight).

Parameter	Amount	Requirements for Adults [110]
Calcium (mg/100 g)	154–426	800 mg
Cupper (mg/100 g)	0.65–0.68	1 mg
Chromium (mg/100 g)	0.94–9.19	40 μg
Iodine (mg/100 g)	41	150 μg
Iron (mg/100 g)	10.99–12.42	14 mg
Magnesium (mg/100 g)	182–204	375 mg
Manganese (mg/100 g)	0.25–1.82	2 mg
Molybdenum (μg/100 g)	38	50 μg
Phosphorus (mg/100 g)	340–528	700 mg
Potassium (g/100 g)	1.34–2.51	2 g
Selenium (mg/100 g)	2.48–37.7	55 μg
Sulfur (g/100 g)	0.35–0.62	-
Zinc (mg/100 g)	5.79–8.25	10 mg
Sodium content (g/100 g)	1.03–1.98	6 g

**Table 5 molecules-27-02300-t005:** Vitamin concentrations in *Yarrowia lipolytica* protein powder (dry weight).

Vitamin	Amount	Requirements for Adults [110]
Vitamin B1 (mg/100 g)	0.68–11.5	1.1 mg
Vitamin B2 (mg/100 g)	1.1–6.9	1.4 mg
Vitamin B6 (mg/100 g)	2.53–6.50	1.4 mg
Vitamin B7 (μg/100 g)	9.32–62.0	50 μg
Vitamin B9 (μg/100 g)	184–330	200 μg
Vitamin B12 (μg/100 g)	5.2–11.2	2.5 μg
Vitamin E (mg/100 g)	0.60–0.75	12 mg

## Data Availability

Not applicable.

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
