# Peer review of "Yarrowia lipolytica* as an Alternative and Valuable Source of Nutritional and Bioactive Compounds for Humans"

_molecules, 2022, doi:10.3390/molecules27072300_

Round 1

Reviewer 1 Report

The manuscript describes recent advances in nutritional and bioactive compounds production using Yarrowia lipolytica. Specifically, Yarrowia lipolytica introduced as an extensive source of protein, exogenous amino acids, unsaturated fatty acids compounds, and vitamins B. The manuscript read well, and provides a great summary of information about nutritional metabolites, genetic and enzymatic production machinery, and conditions for the utilization of wastes and by-products for the production of valuable compounds. Nevertheless, there are several comments regarding the submitted manuscript:

  1. Line 500: The part of “Y. lipolytica as a valuable source of bioactive compounds for food” is too short. Metabolic engineering of lipolytica to produce bioactive compounds seems to be a good choice. Therefore, it is great if the manuscript summarize the engineered Y. lipolytica for the production of valuable compounds.
  2. Please correct the format of references, such as No. 19 (lines 587-588), No. 191 (line 945).

This paper should be accepted for publication after minor revision.

Author Response

Dear Reviewer,

We appreciate Reviewer’s suggestions. They are very valuable for our manuscript.

According to recommendations of Reviewer #1 we revised our manuscript to meet all requirements:

Reviewer’s comments:

  1. Line 500: The part of “Y. lipolytica as a valuable source of bioactive compounds for food” is too short. Metabolic engineering of Y. lipolytica to produce bioactive compounds seems to be a good choice. Therefore, it is great if the manuscript summarize the engineered Y. lipolytica for the production of valuable compounds.

Between lines 503 and 511, we added following paragraph which briefly summarizes metabolic engineering of Y. lipolytica to produce bioactive compounds:

  1. lipolytica is a graceful model to genetically modify to improve its metabolic productivity, increase biomass yields, and/or expand the substrates to be utilized [56, 62, 179, 190]. Metabolic engineering of Y. lipolytica to produce bioactive compounds seems to be a good choice, especially when the yeast is employed to utilize common forestry and agricultural wastes such as cellulose, inulin, starch or xylan, unused by wt strains [77, 191], and/or to produce novel compounds that not produced by the wt yeast. Engineered Y. lipolytica strains produce several heterologous valuable metabolites such as carotenoids [192, 193], terpenes e.g. limonene [194], polyketides with various biological activities and aromatic amino acid-derived molecules [195, 196].

Furthermore, we also added a concluding sentence in Conclusion point, between lines 549 and 551:

Moreover, engineering Y. lipolytica creates new perspectives in terms of its use for production of novel bioactive compounds.

  1. Please correct the format of references, such as No. 19 (lines 587-588), No. 191 (line 945).

We corrected the format of all references according to Author Guidelines.

Reviewer 2 Report

The review is focused on complex evaluation of potential health effects of Yarrowia lipolytica biomass. The review contains an overview of up to date knowledge regarding Y. lipolytica cell composition, cultivation conditions including use of some waste substrates. Moreover, current evaluation of Y-lipolytica safety by European authorities (e.g.EFSA) is introduced and  sufficiently referred.  There are some small mistakes in the text (e.g. in abstract I recommend to use oleaginous yeasts instead oligoneous yeast). Please chceck the text carefully and remove the mistakes.

Author Response

Dear Reviewer,

We appreciate Reviewer’s suggestions. They are very valuable for our manuscript.

According to recommendations of Reviewer #2 we revised our manuscript to meet all requirements:

There are some small mistakes in the text (e.g. in abstract I recommend to use oleaginous yeasts instead oligoneous yeast). Please chceck the text carefully and remove the mistakes.

 We read our manuscript very carefully and removed the mistakes.